# SARS-CoV-2 Consensus-Sequence and Matching Overlapping Peptides Design for COVID19 Immune Studies and Vaccine Development

**DOI:** 10.3390/vaccines8030444

**Published:** 2020-08-06

**Authors:** Alex Olvera, Marc Noguera-Julian, Athina Kilpelainen, Luis Romero-Martín, Julia G. Prado, Christian Brander

**Affiliations:** 1IrsiCaixa AIDS Research Institute-HIVACAT, Hospital Universitari Germans Trias i Pujol, 08916 Badalona, Spain; aolvera@irsicaixa.es (A.O.); mnoguera@irsicaixa.es (M.N.-J.); akilpelainen@irsicaixa.es (A.K.); lromero@irsicaixa.es (L.R.-M.); 2Faculty of Sciences and Technology, Universitat de Vic-Central de Catalunya (UVic-UCC), 08500 Vic, Spain; 3Faculty of Medicine, Universitat de Vic-Central de Catalunya (UVic-UCC), 08500 Vic, Spain; 4Germans Trias i Pujol Research Institute (IGTP), 08196 Barcelona, Spain; 5Institució Catalana de Recerca i Estudis Avançats (ICREA), 08010 Barcelona, Spain

**Keywords:** COVID-19, SARS-CoV-2, consensus sequence, T cell immunity, overlapping peptides set

## Abstract

Synthetic antigens based on consensus sequences that represent circulating viral isolates are sensitive, time saving and cost-effective tools for in vitro immune monitoring and to guide immunogen design. When based on a representative sequence database, such consensus sequences can effectively be used to test immune responses in exposed and infected individuals at the population level. To accelerate immune studies in SARS-CoV-2 infection, we here describe a SARS-CoV-2 2020 consensus sequence (CoV-2-cons) which is based on more than 1700 viral genome entries in NCBI and encompasses all described SARS-CoV-2 open reading frames (ORF), including recently described frame-shifted and length variant ORF. Based on these sequences, we created curated overlapping peptide (OLP) lists containing between 1500 to 3000 peptides of 15 and 18 amino acids in length, overlapping by 10 or 11 residues, as ideal tools for the assessment of SARS-CoV-2-specific T cell immunity. In addition, CoV-2-cons sequence entropy values are presented along with variant sequences to provide increased coverage of the most variable sections of the viral genome. The identification of conserved protein fragments across the coronavirus family and the corresponding OLP facilitate the identification of T cells potentially cross-reactive with related viruses. This new CoV-2-cons sequence, together with the peptides sets, should provide the basis for SARS-CoV-2 antigen synthesis to facilitate comparability between ex-vivo immune analyses and help to accelerate research on SARS-CoV-2 immunity and vaccine development.

## 1. Introduction

Since the start of the COVID-19 pandemic in December 2019, researchers around the world have put major efforts towards a better understanding of the immune response to its causative agent, the SARS-CoV-2. Although an impressive amount of scientific information has been generated in a very short period of time, there remain significant gaps in our understanding of SARS-CoV-2 immune control. In particular, it remains unclear what kind of adaptive immunity should be triggered by vaccination in order to achieve sterile immunity, or at least lead to an ameliorated disease course, in cases where vaccination cannot provide absolute protection from infection. We know from the available literature on other coronaviruses (mainly SARS-CoV-1 and MERS) that antibodies can neutralize the infection, although these humoral responses are short lived in many individuals, and that long-lived T cells responses are present in people with less severe disease outcomes [1,2,3,4,5]. The emerging data on the immune response to SARS-CoV-2 demonstrate the essential contribution of the virus-specific T-cell responses, possibly in addition to the action of neutralizing antibodies, in viral control [3,6,7,8,9,10,11,12,13]. Thus, improved tools to assess host T cell immunity in detail are urgently needed to better identify these responses and to define their role in the outcome of SARS-CoV-2 infection.

Ex-vivo immune analyses of samples from infected individuals can identify T cell responses to specific pathogens like viruses. Such analyses can help to better understand the role of host immunity in virus control and to guide successful vaccine development. However, they rely on the use of the correct recall antigens that can elicit specific responses in vitro. The urgency of the current SARS-CoV-2 pandemic has led researchers to tackle the problem of screening the 10,000 amino acids of the SARS-CoV-2 proteome for T cell responses by selecting viral sequences based on different criteria: (i) bioinformatically predicted epitopes, (ii) homology of SARS-CoV-2 sequences with epitopes defined in other coronaviruses (mainly SARS-CoV) or (iii) selecting some specific SARS-CoV-2 proteins over others [5,7,9,11,14,15,16,17,18,19]. However, all these approaches have intrinsic limitations. Bioinformatic prediction tools are trained on sets of previously described epitopes, but since the available epitope repertoire for many human leukocyte antigen (HLA) alleles is limited, its prediction capacity is also limited [20,21]. Inferences based on epitope sequence homology with other coronaviruses are hampered because past studies on SARS-CoV-1 and MERS only included few selected viral proteins. This is of concern, since screening only a part of the SARS-CoV-2 proteome will potentially miss an important portion of the virus-specific T cell response. Indeed, recent data indicate the existence of T cell responses against structural and non-structural proteins [5,9] for SARS-CoV-2 and other viral infections [22]. Finally, no study has considered the existence of T cell responses to epitopes encoded by open-reading frames (ORF) in alternative frames, as reported for other viral infections [23,24,25,26].

In order to reliably measure total virus-specific T cell immunity, the recall antigens used need to be as representative as possible of the worldwide viral sequences, even for genetically more stable viruses like coronaviruses. T cell recognition of epitopes is very sensitive to mismatches and not matching the recall antigen with the autologous virus can lead to missed responses [27]. For this reason, different test antigen design strategies, trying to cope with the diversity of circulating viral isolates in a single sequence, have been developed in the past. These strategies include central sequence designs such as Center of Tree (COT) [28,29,30,31,32], Ancestral [33,34,35,36] or Consensus sequences [29,30,31,32,35,37,38,39,40,41,42,43]; which may (Ancestral, COT) or may not (Consensus) represent naturally occurring sequences of replication competent viruses. All these designs are sensitive to the underlying sequence database and may change over time as new sequence information on additional isolates becomes available. Direct comparisons of these different central sequence approaches have been performed for a highly variable pathogen (human immunodeficiency virus, HIV) and shown that the different designs yielded comparable results when synthetic peptides covering these sequences were used to measure virus-specific T cell responses [42,43]. However, the additional costs in terms of peptide synthesis and cells needed for ex-vivo experiments, may not warrant inclusion of all the different variants into a single test set.

Thus, the characterization of the complete T cell responses to SARS-CoV-2 urgently needs T cell antigens that cover the whole SARS-CoV-2 proteome while covering sequence diversity, and which can be combined in different experimental set-ups and immune assays. To this end, we created a consensus sequence to cover the genetic diversity of SARS-CoV-2 (CoV-2-cons) for all ORF, including those described in alternative open reading frames. Given the computational ease for its initial generation and periodic updates, we designed a consensus sequence using more than 1700 CoV-2 full-genome sequences and designed overlapping peptide (OLP) sets as recall antigens in T cell assays. The CoV-2-cons OLP sets are presented here in different designs, balancing costs for synthesis with the sensitivity of detecting T cell responses and with the intention to provide a common test antigen that will allow data comparability across laboratories.

## 2. Methods

### 2.1. Consensus Sequence ORF Generation and Entropy Calculation

A total of 1731 full-length SARS-CoV-2 sequences were downloaded from NCBI (30 April 2020, txid2697049, minimum length = 29,000 bp) and aligned using MAFFT [44]. The alignment was visually inspected and curated using Genbank NC_045512.2 as a coordinate reference [45]. A nucleotide consensus sequence was generated by keeping all nucleotides present in at least 25% of the sequences in the alignment. The amino acid consensus sequence was then created by using NC_045512.2 annotated Open Reading Frames (ORFs) plus additional ORFs described in Finkel et al. [46] using the Biostrings R package. Mixed nucleotide positions were either resolved if they were synonymous or flagged for downstream analysis. Positional entropy was calculated at the amino acid level both as the standard and 22-aminoacid-normalized Shannon entropy for every ORF using Bio3d R package on the alignment [47], and afterward, the mean OLP normalized entropy was calculated.

### 2.2. Overlapping Peptide Set Design and Variability Plots

For the automated design of overlapping peptides with variable length, we used the previously described Peptgen algorithm available at the Los Alamos National Laboratories HIV Immunology database [48]. This OLP generator allows predefining peptide length and level of the desired overlap between adjacent OLP. Peptgen is also set up to exclude from the C-terminal end of OLP certain “forbidden” amino acids (G, P, E, D, Q, N, T, S and C) that are rarely seen to serve as the C-terminal anchor position of HLA class I presented epitopes [49]. Using this optional modification can lead to length variation in the OLP set, which can be controlled by limiting the maximal length of an OLP in regions with numerous serial “forbidden” residues. The settings used for the present SARS-CoV-2 consensus OLP design were a) OLP length of 15 or 18 amino acids, with maximal extension or truncation of up to ±3 residues to avoid forbidden C-terminal residues. In addition, the overlap between adjacent OLP was set at 10 or 11 residues. The no-glutamine at N-terminal setting was applied to prevent OLP starting with a glutamine residue as this can lead to complications with peptide synthesis. For positions where two or more amino acids were present above 25% of the sequences in the alignment, two or more sequence variants for those OLPs were generated. Sequence logos were generated for these cases with the ggseqlogo R package [50].

### 2.3. Detection of Conserved Peptides Among Coronavirus

In an attempt to detect protein fragments that are conserved across a wide range of members of the coronavirus family, full-length consensus ORF from SARS-CoV-2 were aligned with other coronavirus sequences. Three alignments were performed based on different sequence selection criteria: (i) 50 reference sequences (RefSeq) with the lowest E-values resulting from a pBLAST search [51] using the ORF-specific consensus sequences (pan-coronavirus alignment) (ii) homologous proteins from 17 viruses representing the Betacoronavirus taxon (beta-coronavirus alignment) or, (iii) homologous proteins from the 7 full-genome sequenced human coronaviruses (including SARS-CoV, MERS-CoV, and common cold species OC43, NL63, 229E, HKU1, human-coronavirus alignment). Selected sequences were aligned using the MUSCLE algorithm in MEGA X [52]. Conserved protein fragments were identified using BioEdit with the following criteria: minimum length of 8 amino acid, maximum average entropy of 0.25, maximum entropy per position of 1 and limiting the search to 1 gap per segment. Sequence logos were generated for the aligned peptides on Weblogo [53].

### 2.4. Identification of Previously Described Epitopes in CoV-2 Conserved Regions

To identify previously reported epitopes in the conserved regions of coronaviruses (pan-coronavirus, betacoronaviruses, and human coronaviruses), and match them with the SARS-CoV-2 consensus sequence, searches for experimentally described epitopes were carried out in the Immune Epitope Database [54]. The search criteria were as follows: “linear peptide; blast option: 90%; Host: Homo sapiens; Any MHC restriction; Positive assays only; All assays; Any disease”. The search yielded 141 epitopes, of which 14 B-cell epitopes and 2 epitopes from a hypothetical protein were removed. The remaining identified epitopes were subsequently used to generate an epitope map of the respective conserved regions.

## 3. Results

### 3.1. Open Reading Frames and Sequence Isolates for CoV-2-Cons Sequence Creation

For creation of the CoV-2 Consensus sequence, nucleotide sequences from 1731 SARS-CoV-2 genomes were aligned and a full genome nucleotide consensus was created, 23 open reading frames (ORF) were then located in the alignment using the NC_045512.2 and the Finkel et al. [46] coordinates and translated to amino acids. Of the 23 ORF, 12 were canonical ORF as annotated in NC_045512.2 and 11 in alternative reading frames described by Finkel et al. [46] (Table 1). In addition, the membrane protein glycoprotein (M), is completely embedded inside an extended ORF (exORFM) without any frameshifts and was not used for separate OLP set design.

### 3.2. Overlapping Peptides (OLP) Sets Design

In order to achieve a balance between the number of peptides needed to cover the whole SARS-CoV-2 proteome, the costs for peptide synthesis and the design of peptide sets that allow for detecting T cell responses with high sensitivity, three OLP sets were designed (Table 2). Shorter peptides (15 mers) with longer sequence overlap between adjacent OLP (11 amino acids) offer high resolution detection of responses, thus lowering the risk of missing longer epitopes located in the OLP overlap. The consequence, however, will be a higher number of peptides to synthesize and screen, in this case a set of 2821 OLP. When the overlap between OLP was reduced from 11 amino acids to 10, the sensitivity of OLP testing is maintained, but some longer epitopes located in the overlap of two OLP may be missed. With this caveat in mind, an OLP set of 15-mers overlapping by 10 residues helped reduce the number of peptides needed by 560 OLP (total number OLP required 2262). Similarly, longer peptides (18 mers) significantly reduce the number of OLP to be synthesized, but tend to reduce in vitro sensitivity [55]. This approach, with an 11 mer overlap, reduced the number of needed OLP to 1561. The final decision for a specific design may also be driven by the assay system used for screening, an a-priori focus on fewer or more viral proteins and the available cells and funding to test immunogenicity. The three full OLP sets with their entropies are included in Appendix A. Of note, the 15–11 OLP sequences were subjected to a search for homologies in the human genome to predict molecular mimicry events related to the autoimmune process. A blastp search (>8aa consecutive identical amino acids per OLP) of the whole set against the human genome yielded no hits.

### 3.3. CoV-2-Cons Variability Analysis by Entropy Scores across the Full Genome

Mismatches between the sequence of in vitro antigen sets and the autologous virus in an infected individual can lead to missed responses. This has been described for highly variable pathogens, such as HCV and HIV, and showed a direct relationship between sequence entropy and the frequency of detected responses [56,57]. Even though the variability of SARS-CoV-2 reported is substantially lower than for HIV and HCV, the sequence entropy was calculated at the amino acid level and as the mean OLP entropy in order to identify positions and OLP that may escape detection in T cell screening assays.

Amino acid positional Shannon entropies were generally highly conserved, although specific more variable positions were identified (Appendix A), linked to specific amino acid variants. The ORF1ab protein, including three of the most variable positions, is shown in Figure 1. In the CoV-2-cons 15–11 OLP set, mean OLP normalized entropies were overall low (Range: 0.947–0.758) and comparable between OLP covering the canonical ORF (Range: 0.947–0.879) and OLP matching the alternative frameshift ORF (Range: 0.932–0.758).

### 3.4. Variant OLP Sequences to Cover CoV-2 Sequence Diversity

Based on the SARS-CoV-2 alignment used to design the consensus, only nine amino acid positions in the entire SARS-CoV-2 genome showed two amino acids present in at least 25% of the sequences (Figure 2). Three of them were located in ORF1ab, one in the RNA polymerase and two in the Helicase sub-proteins. None of them were located close enough to each other to affect the same OLP. Still, the synthesis of a single consensus peptide could miss T cell responses in individuals exposed to the virus with the subdominant sequence variant. To prevent missing responses, a small number of additional OLP containing each of the variants were generated to cover the variability of these OLP, creating an additional set of 31 different variant OLP in the 15–11 OLP set (Table 2).

### 3.5. Conserved Protein Sequences Matching Other Coronavirus Family Member and Identification of Pan-Coronavirus Sequences

In addition to variable positions, we also evaluated the presence of protein regions conserved among coronavirus species, as these may support the design of immunogen sequences for pan-coronavirus vaccines. A total of 26 regions, ranging from 8 to 23 amino acids, were identified as being conserved in at least one of the three different sequence alignments (Table 3). Fifteen fragments were identified in the pan-coronavirus alignment, 17 in the beta-coronavirus alignment and 12 in the human coronavirus alignment. Seven of them were detected in all three alignments. To identify potential T cell epitopes in these conserved regions, we searched the IEDB for described T-cell epitopes similar (>90% sequence identity) to the conserved peptides present in the CoV-2 consensus sequence. Interestingly, the majority of the conserved regions contained several matches, most of which were described epitopes derived from SARS-CoV. In total, 125 similar epitopes were identified, from all but two of the conserved regions (Table 3). The similar epitopes were found to be derived from the following organisms; SARS-CoV: 71, Human coronavirus 229E: 1, Alphacoronavirus 1: 1, Unknown origin: 3, and Homo sapiens: 47. Interestingly, 24 out of 26 fragments contained the described SARS-CoV T cell epitopes, indicating that these regions are immunogenic in humans and reinforcing the idea that some degree of cross-reactivity among coronavirus can be expected [11,58]. Also, the majority, i.e., 40 of the 47 human epitopes, clustered around one single region conserved in the beta-coronavirus alignment (QGPPGTGKSH). Several conserved peptides have thus been identified, which could potentially contain epitopes cross-reactive among different Coronavirus species. These conserved peptides can thus provide valuable information to understand if the immune response to SARS-CoV-2 is affected by previous infection with other coronaviruses and for pan-coronavirus vaccine design (Appendix A).

## 4. Discussion

We here report the design of a CoV-2-cons sequence and the matched OLP sets for the comprehensive analysis of the adaptive T cell immune response against SARS-CoV-2. Three sets of OLP reported here provide enough flexibility to balance exhaustive screening for T cell responses and available resources. Ideally, the wide use of such a CoV-2-cons sequence and a specific OLP set (ideally 15 mer with 11 overlap) would ensure the comparability and reproducibility of immunological data across laboratories worldwide to accelerate SARS-CoV-2 immunological studies.

Fifteen-mer designs allow sensitive screens for both, CD4+ and CD8+ T cell responses while 18 mer allow for cheaper peptide synthesis and require less cells for comprehensive screenings. However, longer test peptides tend to yield fewer responses and imply bigger efforts for subsequent epitope mapping. For the 15 mer design, an alternative 10 amino acid overlap was proposed to reduce peptide synthesis, while maintaining the sensitivity. This approach may be valuable, but may miss epitopes restricted by HLA class I molecules known to presented longer peptides (such as HLA-B*27, -B*57 and others). Regardless of the final OLP design, the use of large OLP data sets for immune screening raises several challenges. How to pool peptides in suitable numbers may depend on the downstream analyses, whether or not subsequent epitope identification are planned, on the experimental setup and whether long incubation periods will be required. The latter may be especially important as pooling of a large number of peptides will possibly require lyophilization of the pooled peptides to eliminate dimethyl sulfoxide (DMSO) as this can be toxic for the cells during culture [11]. Also, as we gain more insights into the distribution of virus-specific T cell responses across the full proteome, more or less reactive regions can be pooled based on expected reactivity, protein expression level, and/or degree of conservation [46].

Canonical and alternative frame ORF were considered in the present CoV-2-consensus sequence design to ensure an as broad as possible screening for all potentially expressed protein sequences. Whether all these putative ORF are indeed expressed remains to be confirmed. If shown that not all these sequences are indeed expressed, the OLP set could be reduced by some 65 peptides, focusing exclusively on the canonical ORF. Consensus sequence design is highly dependent on the sequences included in the alignments used to construct them. We used publicly available sequences in the growing SARS-CoV-2 NCBI repository as a representative set of worldwide sequences. As noted, coverage of sequence diversity for in-vitro antigen test sets is critical as responses to autologous viral variants may be missed if these variant sequences are not matched [27]. This may be most critical for highly variable pathogens, such as HCV and HIV, where it has been shown that sequence entropy was directly related to the frequency of OLP reactivity in vitro and essential to identify the potential emergence of immune escape variants [59,60]. However, even genetically more stable pathogens such DNA viruses (for instance Epstein Barr Virus, EBV) have been reported to exist as a swarm of quasi-species and to lose specific T cell epitopes over time [61,62]. This is also supported by recent data showing some degree of adaptation to host immunity and sequence variability for SARS-CoV-2 as it moves through the global human population [63]. To cover these variant sites, variant OLP can be synthesized. An alternative approach to the synthesis of individual variant peptide sequences is the use of “toggled peptides”, where the sequence variation is directly incorporated into the peptide synthesis. To achieve this, peptide synthesis uses mixes of amino acids at variable positions, so that the resulting OLP resembles a mini-peptide library that can achieve an a-priori set coverage of circulating viral variants [64]. This would readily allow to cover more sequence diversity beyond the 25% frequency cut-off that was applied in the present study.

The existence of protein fragments conserved among different coronavirus species has several implications. For the interpretation of T cell responses, it has to be taken into account that some degree of cross-reactivity can exist among human coronavirus [5,65]. This implies that responses to these regions could be associated with previous infections by other human coronaviruses, some of them triggering much milder infections that can pass unnoticed, like those by coronaviruses causing a common cold. This observation will need to be taken into consideration when interpreting immune data on SARS-CoV-2. On the other hand, the existence of conserved sequences among beta- or even the whole coronavirus family suggests that T cell responses to these regions could provide broad protection and that the creation of a pan-coronavirus vaccine may be feasible. Such a vaccine could allow to prevent infection not only with SARS-CoV-2, but also with other, clinically relevant coronavirus like SARS-CoV-1 and MERS, and even with new coronaviruses jumping the species barrier to humans. However, the design of a pan-coronavirus vaccine will critically depend on the identification of epitopes shared among them. These pan-coronavirus epitopes are likely to exist in conserved sequences, but need to be experimentally validated. At the same time, the existence of SARS-CoV-2 homologous regions in the human genome, together with the existence of described epitopes in these regions raise some concern that coronaviruses could be involved in a molecular mimicry process triggering autoimmune diseases like the Guillain-Barré syndrome [66,67,68,69].

The present study is currently limited to the design of the CoV-2 consensus sequence, without functional immune analyses of the OLP sets in samples from infected individuals. However, the principal aim here was to provide a SARS-CoV-2 T cell test reagent, including all described ORF and covering as much viral variability as possible, for its implementation in future screening efforts. In addition, the OLP sets will certainly elicit T cell responses in vitro as partial evaluation has been performed by others in studies using peptides spanning some of the regions covered by the present consensus sequence [5,9,11] and since the current peptide designs (length, overlap) has been shown to be effective in the past [55,70]. Thus, the present peptide designs will afford a high-resolution analysis of the T cell response to SARS-CoV-2, the nature of the targeted epitopes and the functionality and T cell receptor use of the T cells targeting these epitopes, thereby increasing our knowledge of factors that drive COVID-19 disease progression and which could be implemented in vaccine development.

## 5. Conclusions

We here present the first SARS-CoV-2 Consensus sequence for all described SARS-CoV-2 ORF, including those in alternative frames covering the SARS-CoV-2 sequence variability represented by 1700 available sequences. The description of this sequence and of the matching OLP sets will aid the further immune analyses in SARS-CoV-2 infection and ensure reproducibility between laboratories. In light of recent studies, the T cell response to SARS-CoV-2 can be crucial to control SARS-CoV-2 infection. To date, published studies are generally limited to a few viral proteins, using recall antigens that do not reflect sequence diversity nor alternative ORFs. To overcome these limitations, the description of the global landscape of T cell responses to SARS-CoV-2 urgently needs unbiased, comparable, full-proteome screens for virus-specific T cell responses. The CoV-2-cons and matched OLP sets described here will allow to integrate data globally, generating crucial information for vaccine development. We also include measures of sequence entropy to identify the most variable segments and design additional OLP sequences that cover these sites. Of note, these entropy analyses, together with sequence alignments across a wide range of coronaviruses, also allowed the identification of highly conserved regions among different coronaviruses. These regions may be targeted by T cells, which could target a wide range of coronaviruses and may be relevant targets for T cell vaccine design.

## Figures and Tables

**Figure 1 vaccines-08-00444-f001:**
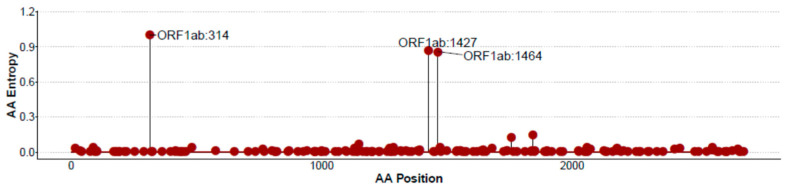
Standard Shannon entropy plot by amino acid position for ORF1ab. Zero entropy indicates total conservation at each specific position.

**Figure 2 vaccines-08-00444-f002:**
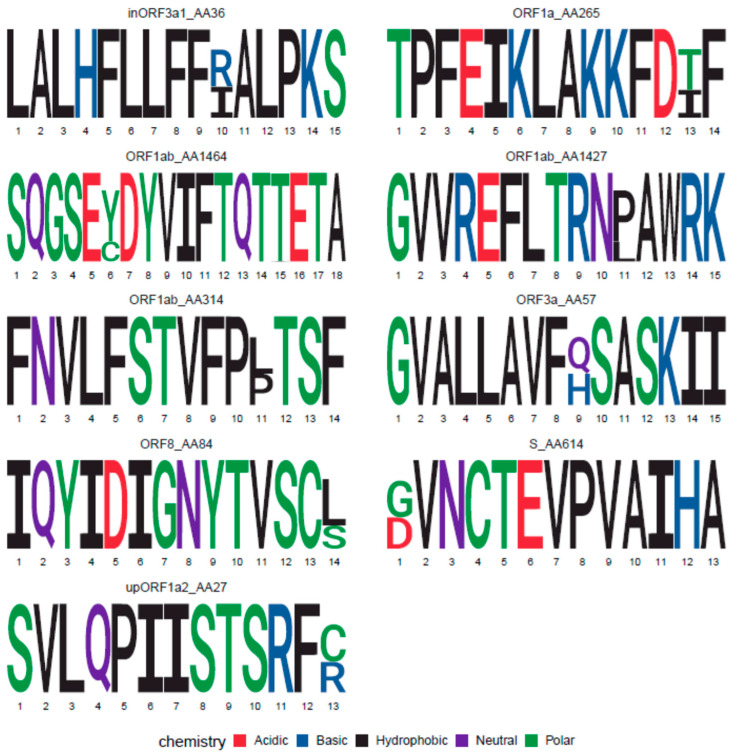
Sequence Logos for epitopes encompassing variable (>25%) positions. Protein location and starting amino acid positions are indicated on top of the logo.

**Table 1 vaccines-08-00444-t001:** Canonical and alternative open reading frames (ORF) in SARS-CoV-2. iORF: internal OPF, extORF: extended ORF, upORF: upstream ORF.

Gene	Start	End	Protein	Protease Products	Frame
ORF1a.iORF1.ext	59	136	upORF1a1	-	Alternative
ORF1a.iORF2.ext	163	264	upORF1a2	-	Alternative
ORF1ab	266	13483	pp1a	leader protein	Canonical
				nsp2	
				nsp3	
				nsp4	
				3C-like proteinase	
				nsp6	
				nsp7	
				nsp8	
				nsp9	
				nsp10	
				nsp11	
ORF1ab	13468	21555	pp1ab	RNA-dependent RNA polymerase	Canonical
				helicase	
				3′-to-5′ exonuclease	
				endoRNAse	
				2′-O-ribose methyltransferase	
S	21563	25384	surface glycoprotein	S1	Canonical
				S2	
ORFS.iORF1	21744	21863	inORFS	-	Alternative
ORF3a	25393	26220	ORF3a protein	-	Canonical
ORF3a.iORF1	25457	25582	inORF3a1	-	Alternative
ORF3a.iORF2	25596	25697	inORF3a2	-	Alternative
E	26245	26472	envelope protein	-	Canonical
ORFM.ext	26484	27191	exORFM	-	Alternative
M	26523	27191	membrane glycoprotein	-	Canonical
ORFM.iORF	27151	27195	inORFM	-	Alternative
ORF6	27202	27387	ORF6 protein	-	Canonical
ORF7a	27394	27759	ORF7a protein	-	Canonical
ORF7b	27756	27887	ORF7b protein	-	Canonical
ORF7b.iORF2	27862	27897	inORF7b	-	Alternative
ORF8	27894	28259	ORF8 protein	-	Canonical
ORF8.iORF	27965	27994	inORF8	-	Alternative
N	28274	29533	nucleocapsid phosphoprotein	-	Canonical
ORFN.iORF1	28284	28577	ORF9b	-	Alternative
ORF10.upORF	29538	29570	upORF10	-	Alternative
ORF10	29558	29674	ORF10 protein	-	Canonical

ORF position is referred to the NC_045512.2 reference sequence.

**Table 2 vaccines-08-00444-t002:** Description of the three CoV-2 OLP sets.

Set	Length	Overlapp	Number	Variants
15–11	15	11	2821	31
15–10	15	10	2262	23
18–11	18	11	1561	22

**Table 3 vaccines-08-00444-t003:** Conserved sequences among different coronavirus. I: Pan-coronavirus, II: Betacoronavirus, III: Human coronavirus alignment. The black squares that indicted which alignments contained the conserved sequences.

Consensus Sequence	ORF	Consensus Start Position	Alignment Hit			Epitopes			
			I	II	III	Unknown	SARS-CoV	Human	Other Coronavirus
VGVLTLDNQDLNG	ORF1b	193				1	4	-	-
TQMNLKYAISAKNRARTVAGVSI	ORF1b	530				-	5	2	-
VIGTSKFYGGW	ORF1b	580				-	3	-	-
LMGWDYPKCDRAMPN	ORF1b	605				1	3	-	-
LANECAQVL	ORF1b	646				-	1	-	-
YVKPGGTSSGDATTA	ORF1b	665				-	3	-	-
KHFSMMILSDDAVVCFN	ORF1b	743				-	2	1	-
LYYQNNVFMS	ORF1b	778				-	-	-	-
GPHEFCSQHT	ORF1b	800				-	2	-	-
LPYPDPSRIL	ORF1b	820				-	2	3	-
ERFVSLAIDAYPL	ORF1b	849				-	5	-	1
SQTSLRCG	ORF1b	934				-	1	-	-
LYLGGMSYY	ORF1b	986				-	3	-	-
LKLFAAET	ORF1b	1054				-	4	-	-
QGPPGTGKSH	ORF1b	1205				1	2	40	-
TACSHAAVDALCEKA	ORF1b	1231				-	1	-	-
GDPAQLPAPR	ORF1b	1324				-	3	-	-
AVFISPYNSQN	ORF1b	1432				-	4	1	-
NRFNVAITRA	ORF1b	1483				-	2	-	-
CNLGGAVC	ORF1b	2002				-	1	-	-
KYTQLCQYLN	ORF1b	2443				-	3	-	-
RSFIEDLLF	Spike	815				-	2	-	-
QIDRLITGRL	Spike	993				-	5	-	1
KWPWYIWL	Spike	1211				-	-	-	-
WSFNPETN	M	110				-	3	-	-
PRWYFYYLGTGP	N	106				-	7	-	-

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
