# Peer review of "SARS-CoV-2 Consensus-Sequence and Matching Overlapping Peptides Design for COVID19 Immune Studies and Vaccine Development"

_vaccines, 2020, doi:10.3390/vaccines8030444_

Round 1
Reviewer 1 Report
This manuscript describes the design of overlapping peptides (OLP) to study the T cell response against SARS-CoV-2. First, the authors generated a consensus sequence of SARS-CoV-2 from over 1,700 viral genomes. Using this consensus sequence, the authors identified OLP sets of various peptide lengths and sequence overlap ultimately listing three sets between 1561 and 2821 OLPs to cover all of the open reading frames. Additional OLPs were suggested based on non-conserved regions of the SARS-CoV-2 genome. Finally, the authors identified conserved regions across coronaviruses that might be cross-reactive epitopes.
This manuscript identified multiple peptide libraries that could be useful for future SAR-CoV-2 studies analyzing T cell immunity. However, this manuscript only describes the peptides rather than functionally testing the peptides.
Major comments
- The authors need to test some of the peptides for T cell stimulation. While testing >1561 peptides is an enormous task in both time and finances, some experimental validation of the peptides to stimulate T cells is needed.
- Currently, the manuscript does not add much to the SARS-CoV-2 field. The authors aligned sequences and used a program to design overlapping peptides for ex vivo immune analysis, but did not perform any immune analysis. However, the authors did not take the next step of testing any of the potential T cell peptides.
Minor comments
- Additional citations are needed for lines 53 and 68.
- The authors cite a BioRxiv paper that evaluated T cell responses from SARS-CoV-2 recovered patients (Bert et al, 2020). Were any of the epitopes identified in that manuscript present in your OLP panels?
Author Response
- This manuscript identified multiple peptide libraries that could be useful for future SAR-CoV-2 studies analyzing T cell immunity. However, this manuscript only describes the peptides rather than functionally testing the peptides.
Author’s response: The aim of our study is to provide the field with urgently needed information on consensus sequence alignment and peptide set libraries that can be widely used by the CoV-2 research community for comprehensive, full proteome screens of T cell responses to SARS-CoV-2. We feel that it is critical to prioritize publication of these sequences and peptide sets to overcome current limitations of CoV-2 T cell studies, which at this time are generally based on “in silico” predicted epitopes, are limited to selected CoV-2 proteins, do not take into account alternative frame ORF and do not deal with global sequence variability. Thus, the use of in vitro recall antigens that are biased or incomplete, can miss CoV-2 epitopes that could be crucial to understand the adaptive immune response to the virus and for vaccine design. In the revised discussion, we address this reviewer´s concern and highlight that the use of the same OLP sets across laboratories will improve immune data reproducibility and integration of anti-CoV-2 T cell response data to accelerate and guide vaccine design around the world.
- The authors need to test some of the peptides for T cell stimulation. While testing >1561 peptides is an enormous task in both time and finances, some experimental validation of the peptides to stimulate T cells is needed.
Author’s response: We agree with the request for a functional test of the OLP sets to determine its value as a recall antigen. Based on recent published data by Le Bert et al 2020 we already know that some of these peptides are immunogenic. However, we believe that adding few functional data that could be generated in the short term, and without unduly delaying the publication of these globally usable peptide sequences will not add substantially important data. We have ongoing efforts in place for peptide matrix design and peptide production to start T cell screens. We have added a paragraph (lines 299-309) of study limitations in the discussion section.
- Currently, the manuscript does not add much to the SARS-CoV-2 field. The authors aligned sequences and used a program to design overlapping peptides for ex vivo immune analysis, but did not perform any immune analysis. However, the authors did not take the next step of testing any of the potential T cell peptides.
Author’s response: As we stated in the previous questions the primary aim of the study was the design of a consensus sequence for CoV-2 and the corresponding peptide sets. The design of a consensus sequences and OLP sets is not trivial, but tedious and labor intensive. Thus, we great value having this information readily available for the CoV-2 researchers. A small T cell functional screen will not add value to the current study if we consider data generated in other laboratories that tested some peptides that overlapped with some of the regions covered by our peptide sets (Le Bert et al 2020). We have added these references in the manuscript. Form the work by Le Bert and colleagues, it is also clear that the assessment of global T cell responses to CoV-2 in an unbiased and full proteome screen is needed. In addition, we want to stress that using the same OLP set by different laboratories should increase the comparability of the data, allowing data integration to speed vaccine development. As mentioned before, the need of such a tool argue for urgent publication.
In response to the reviewers´ comments, we have now rewritten the introduction to clarify the aim of this study and highlight the added value of the consensus sequence and the OLP in the context of current data. Also, we have introduced a paragraph (lines 308-319) in the discussion section for study limitations, and, as requested by the editors, added a conclusion section that clearly highlights these points.
- Additional citations are needed for lines 53 and 68.
Author’s response: We have added the needed references to lines 53 and 68
- The authors cite a BioRxiv paper that evaluated T cell responses from SARS-CoV-2 recovered patients (Bert et al, 2020). Were any of the epitopes identified in that manuscript present in your OLP panels?
Author’s response: Yes, they are all present in our OLP panels as mentioned in previous answers
Reviewer 2 Report
Since the start of pandemic by SARS-CoV-2, the scientist all over the world are trying their best to address the menace from different angles with the aim of increasing our existing knowledge about the SARS-CoV-2virus. The authors in the present study have aligned the available sequence data for SAR-Cov-2 and identified conserved regions in the current circulating strains of SARS-CoV 2 as well as pan-CoV sequences. In general, the study is presented nicely. However, some minor points need to be addressed.
- The authors should discuss the detection of antibodies against internal proteins of some viruses such as influenza virus (Carragher D. M et al., J Immunol. 2008) to support the idea of current study.
- Page 4, Lines 146-148: Earlier in materials and methods the authors stated that more than 1700 sequences were aligned to create consensus sequence. Here, the authors stated that “nucleotide sequences from 23 open reading frames (ORF) were aligned and then translated to amino acids consensus sequences”. This statement needs clarification since the procedure for generation of consensus amino acid sequence seems to be different from the procedure described here i.e. why did the authors derive 23 ORFs?
Author Response
- The authors should discuss the detection of antibodies against internal proteins of some viruses such as influenza virus (Carragher D. M et al., J Immunol. 2008) to support the idea of current study.
Author’s response: The study by Carragher and others in this regard, have reported Ab responses to internal proteins of Influenza virus and underlying T cell reactivity to drive this humoral immunity. We have included this concept in the introduction (lines 64-65) as we fully agree that these antibody responses may be induced in vivo and sustained by specific T cell responses to epitope derived from these proteins. This is a further argument to conduct comprehensive T cell screenings as we propose them here rather than limiting these studies to the usually tested (surface) antigens.
- Page 4, Lines 146-148: Earlier in materials and methods the authors stated that more than 1700 sequences were aligned to create consensus sequence. Here, the authors stated that “nucleotide sequences from 23 open reading frames (ORF) were aligned and then translated to amino acids consensus sequences”. This statement needs clarification since the procedure for generation of consensus amino acid sequence seems to be different from the procedure described here i.e. why did the authors derive 23 ORFs?
Author’s response: We thank the reviewer for bringing this point to our attention and have better described the process of creating the ORF consensus sequences in the results section.
Reviewer 3 Report
Work is well presented and described. One edit change and one suggestion:
- on p.8 line 266 replace SARS-CoV2 to SARS-CoV-2 for consistency;
- in description for Fig. 1 spell-out abbreviations for iORF, ext, uORF, upORF.
The manuscript describes creation of SARS-CoV-2 consensus sequence created based on 1700 viral genomes and including all canonical and alternative open reading frames of the virus. Based on the consensus sequence the authors created 3 overlapping peptide lists which contain between 1500 to 3000 peptides, depending on the selection criteria. Conserved among different coronaviruses and variable peptides were identified and described and alternative peptides were added for variable regions of the putative proteins. Generated consensus sequence and peptide lists can serve as a reference and basis for antigen synthesis for immune-analysis for development of new generation vaccines.
The manuscript is well written and data are well described and presented. This work will be of great benefit to research community, especially for the researches working on SARS-CoV-2 vaccines, especially those targeting T-cell mediated responses. Corrections and suggestions to the authors are minimal and presented in the comments to the authors.
Author Response
- on p.8 line 266 replace SARS-CoV2 to SARS-CoV-2 for consistency
Author’s response: we have searched “SARS-CoV2” through the text and changed for “SARS-CoV-2” for consistency as suggested by the reviewer.
- in description for Fig. 1 spell-out abbreviations for iORF, ext, uORF, upORF.
Author’s response: We have added the following sentence to Table 1 legend: “iORF: internal OPF, extORF: extended ORF, upORF: upstream ORF” The abbreviation “uORF” has been changed in the table for “upORF” for consistency.
Reviewer 4 Report
In this manuscript by Olvera and colleagues, a consensus sequence for SARS-CoV-2 is described and investigated. Therefore, the authors included more than 1700 sequenced SARS-CoV-2 genomes including newly identified ORFs.
The work can be easily divided into two parts: The first segment encompasses variability analysis and sequence diversity in overlapping peptides created from the described SARS-CoV-2 consensus sequence. In the second part, the described SARS-CoV-2 consensus sequence was aligned with other coronavirus sequences (Pan-Coronavirus, Betacoronavirus, human coronavirus) to identify conserved sequences and epitopes.
The article shows a clear and precise approach, however the novelty and relevance of the findings needs to be pointed out more clearly.
Minor points:
The Authors should stress how their approach differs from those already published SARS-CoV-2 consensus sequences and overlapping peptide lists. In the light of vaccine development it needs to be more emphasized, why a pan-Coronavirus vaccine would be of relevance and what might be the caveats of such a vaccine. Furthermore, the known characteristics of T-cell memory responses to other coronaviruses should at least be discussed in relation to this topic.
The introduction is too long and the relevance of paragraph 3 (lines 62-81) for the authors work has to be pointed out more clearly.
Figure 1: The labels of the second and third position overlap and are therefore hard to decipher. Furthermore, the frame is incomplete and the top label of y-axis is cut.
Figure 2 needs a better labelling, e.g. ORFs or positions of the variable positions, and a more comprehensible legend.
Author Response
- The article shows a clear and precise approach; however, the novelty and relevance of the findings needs to be pointed out more clearly.
Author’s response: We have now clarified the relevance of the present publication by rewriting parts of the introduction and discussion sections. Also, as requested by the editor, we have added a conclusions section to highlight the impact of the study.
- The Authors should stress how their approach differs from those already published SARS-CoV-2 consensus sequences and overlapping peptide lists. In the light of vaccine development, it needs to be more emphasized, why a pan-Coronavirus vaccine would be of relevance and what might be the caveats of such a vaccine. Furthermore, the known characteristics of T-cell memory responses to other coronaviruses should at least be discussed in relation to this topic.
Author’s response: We have modified the discussion and the introduction according to the reviewer suggestions. In particular, we have: i) rewritten the introduction (Lines 53-64) to compare our approach with previous strategies for antigen design, ii) discussed the relevance and caveats of a pan-coronavirus vaccine in the discussion (Lines 290-295), and iii) added in the introduction a brief description (lines 42-45) of previous data on responses to other coronavirus and how they relate to SARS-CoV-2 responses.
- The introduction is too long and the relevance of paragraph 3 (lines 62-81) for the authors work has to be pointed out more clearly.
Author’s response: We have rewritten the third paragraph of the introduction to emphasize the importance of dealing with sequence variability and the whole proteome for comprehensive understanding of the T cell CoV-2 landscape. As requested by the reviewer, we have also shortened the introduction.
- Figure 1, the labels of the second and third position overlap and are therefore hard to decipher. Furthermore, the frame is incomplete and the top label of y-axis is cut.
Author’s response: We have changed Figure 1 to correct these issues
- Figure 2 needs a better labelling, e.g. ORFs or positions of the variable positions, and a more comprehensible legend.
Author’s response: We have changed Figure 2 to improve the labelling and the legend
Round 2
Reviewer 1 Report
This is a revised version of a manuscript describing the design of overlapping peptides to study SARS-CoV-2 T cell immunity and potential cross-reactivity between other coronaviruses. I appreciate the additional text that was added clarifying the aims of the paper, identifying the limitations to the study and the need to validate the identified peptides. With these modifications, the manuscript should be accepted.
Minor comment
1- The reference (Finkel et al) is misspelled in line 165.
Author Response
We have corrected the misspelling detected by reviewer 1.